# Confidence in Women’s Health: A Cross Border Survey of Adult Nephrologists

**DOI:** 10.3390/jcm8020176

**Published:** 2019-02-03

**Authors:** Elizabeth M. Hendren, Monica L. Reynolds, Laura H. Mariani, Jarcy Zee, Michelle M. O’Shaughnessy, Andrea L. Oliverio, Nicholas W. Moore, Peg Hill-Callahan, Dana V. Rizk, Salem Almanni, Katherine E. Twombley, Emily Herreshoff, Carla M. Nester, Michelle A. Hladunewich

**Affiliations:** 1Division of Nephrology, Department of Medicine, Sunnybrook Health Sciences Centre, University of Toronto, Toronto, ON M4N 3M5, Canada; ehendren@gmail.com; 2Division of Nephrology, Department of Internal Medicine, University of North Carolina, Chapel Hill, NC 27599, USA; Monica.Reynolds@unchealth.unc.edu; 3Division of Nephrology, Department of Internal Medicine, University of Michigan, Ann Arbor, MI 48109, USA; lmariani@med.umich.edu (L.H.M.); aoliv@med.umich.edu (A.L.O.); 4Arbor Research Collaborative for Health, Ann Arbor, MI 48104, USA; Jarcy.Zee@arborresearch.org (J.Z.); Nick.Moore@arborresearch.org (N.W.M.); peghillcallahan@comcast.net (P.H.-C.); 5Division of Nephrology, Department of Medicine, Stanford University School of Medicine, Palo Alto, CA 94305-5101, USA; moshaugh@stanford.edu; 6Division of Nephrology, Department of Medicine, University of Alabama at Birmingham, Birmingham, AL 35294, USA; drizk@uabmc.edu; 7Division of Nephrology, Department of Medicine, Ohio State University Wexner Medical Center, Columbus, OH 43210, USA; Salem.Almaani@osumc.edu; 8Division of Nephrology, Department of Pediatric, Medical University of South Carolina, Charleston, SC 29425, USA; twombley@musc.edu; 9Division of Nephrology, Department of Pediatrics and Communicable Diseases, University of Michigan, Ann Arbor, MI 48109, USA; egalopin@med.umich.edu; 10Divisions of Nephrology, Departments of Internal Medicine and Pediatrics, University of Iowa, Iowa City, IA 52242, USA; carla-nester@uiowa.edu

**Keywords:** surveys and questionnaires, glomerular disease, chronic kidney disease, pregnancy outcome, pregnancy complication

## Abstract

A range of women’s health issues are intimately related to chronic kidney disease, yet nephrologists’ confidence in counseling or managing these issues has not been evaluated. The women’s health working group of Cure Glomerulonephropathy (CureGN), an international prospective cohort study of glomerular disease, sought to assess adult nephrologists’ training in, exposure to, and confidence in managing women’s health. A 25-item electronic questionnaire was disseminated in the United States (US) and Canada via CureGN and Canadian Society of Nephrology email networks and the American Society of Nephrology Kidney News. Response frequencies were summarized using descriptive statistics. Responses were compared across provider age, gender, country of practice, and years in practice using Pearson’s chi-squared test or Fisher’s exact test. Among 154 respondents, 53% were women, 58% practiced in the US, 77% practiced in an academic setting, and the median age was 41–45 years. Over 65% of respondents lacked confidence in women’s health issues, including menstrual disorders, preconception counseling, pregnancy management, and menopause. Most provided contraception or preconception counseling to less than one woman per month, on average. Only 12% had access to interdisciplinary pregnancy clinics. Finally, 89% felt that interdisciplinary guidelines and/or continuing education seminars would improve knowledge. Participants lacked confidence in both counseling and managing women’s health. Innovative approaches are warranted to improve the care of women with kidney disease and might include the expansion of interdisciplinary clinics, the development of case-based teaching materials, and interdisciplinary treatment guidelines focused on this patient group.

## 1. Introduction

Across their lifespan, women with chronic kidney disease (CKD) face unique challenges and require comprehensive care [1]. In women with CKD, hormonal contraception can raise blood pressure, worsen proteinuria, and may exacerbate osteopenia [2,3,4]. Medications routinely used to treat kidney disease can be teratogenic or reduce fertility. Menstrual irregularities, including abnormal uterine bleeding and anovulation, become common as kidney disease progresses [5]. Women with end-stage renal disease (ESRD) have alterations in the hypothalamic-gonadal axis, provoking functional menopause [6]. Risks for fracture and osteoporosis are also significantly higher in older women with CKD [5].

When able to conceive, women with CKD also experience high rates of maternal and fetal complications. A systematic review and meta-analysis of pregnancy outcomes in women with CKD found an approximately 10-fold increased risk for preeclampsia, a 5-fold increased risk for preterm delivery and small for gestational age infants, and an almost 3-fold risk for cesarean delivery [7]. These risks appear to increase in a step-wise fashion with advancing CKD stage and severity of proteinuria [8]. Despite what is known about pregnancy in patients with CKD, women experience variability in messages conveyed by nephrologists concerning fertility, family planning, and the risk of pregnancy to themselves and their offspring [9]. In qualitative studies, some women reported feeling traumatized when their physician aggressively warned against pregnancy [10]. Others expressed insecurities surrounding their health and disease progression as well as feelings of guilt, body failure, and grief [10,11].

Nephrologists are uniquely positioned to provide disease-specific family planning advice to women of child-bearing age with CKD. When asked, most women with chronic diseases feel that their specialists should be initiating discussions regarding preconception care [12]. However, these issues have not been consistently addressed by nephrologists [11,13]. A survey of nephrologists and allied health professionals in Canada, Australia, and New Zealand focusing on menstrual disorders in women with CKD showed a minority (35%) reported discussing fertility with their patients, and even fewer (15%) discussed menstrual irregularities [13]. In a survey of German nephrologists regarding pregnancy in women on dialysis, only 45% of respondents reported routinely educating their dialysis patients of child-bearing age about contraception [14]. Finally, a survey of women with CKD revealed that only half were offered contraception counseling, and less than a quarter were offered preconception counseling [11].

While prior studies have demonstrated low rates of counseling in women’s health, nephrologists’ confidence in counseling or managing these issues has not been fully addressed. By uncovering areas of low confidence, targeted strategies for improved care can be developed. Through an electronic survey, the women’s health working group of Cure Glomerulonephropathy (CureGN) aimed to evaluate exposure to women’s health issues in training and current practice, confidence in counseling and managing these issues, and ways to improve the future care of women with CKD.

## 2. Materials and Methods

### 2.1. Survey Design and Content

Figure 1 presents a flowchart of the survey development and dissemination. Survey question construction was grounded in a comprehensive literature review of qualitative and quantitative studies of women’s health in CKD [9,10,11,12,13,14]. Three main themes arose: participant characteristics and fellowship training, current clinical practices, and confidence levels in counseling and managing women’s health. Through recurring conference calls, the CureGN women’s health working group, which consists of experts in women’s health and CKD (n = 24), developed each item of the survey and refined wording, content, and relevance with each iteration. The electronic version was designed and constructed using the SurveyMonkey platform (SurveyMonkey Inc. San Mateo, California, USA) [15]. This pilot survey was then tested by six experts in kidney disease or women’s health (not part of the CureGN women’s health working group) to confirm face and content validity as well as survey functionality.

The final 25-item survey can be found in the Appendix A. Collected demographic information included age, gender, country where nephrology training was completed, country of current residence, years in practice, practice setting, clinical scope of practice, and percentage of time dedicated to clinical care. Fellowship training in obstetric nephrology and/or women’s health was inquired about as well as the number of pregnant women seen during training. We also asked respondents to report the average number of women per month they counseled on contraception/family planning and preconception care in the past 12 months. We asked respondents to report how many pregnant women with kidney disease they cared for in the past 12 months, how many pregnant women with a kidney transplant they cared for in the past five years, and how many pregnant women on dialysis they cared for in their career. A free-text follow-up question regarding dialysis prescribing practices in pregnancy was posed.

Clinical documentation practices of the obstetric history were assessed using a Likert frequency scale (never, sometimes, rarely, often, or always). A Likert scale was also employed to assess provider confidence (not at all confident, somewhat confident, confident, or very confident) in general women’s health topics and pregnancy-specific issues. To evaluate barriers to optimal counseling and to uncover strategies that might improve future care, we asked respondents what limits their ability to provide reproductive/obstetric counseling and what resources might help them to better counsel and manage women with CKD.

### 2.2. Study Sample and Dissemination

The study was reviewed by the University of North Carolina Office of Human Research Ethics and was determined to be exempt from requiring participant consent due to the nature of the anonymous survey. The electronic survey was distributed from February to April 2018 (see Figure 1). The targeted participants were adult nephrologists, both in training and in practice. A convenience sample was obtained from email dissemination to the Canadian Society of Nephrology (CSN) (*n* = 377) and the CureGN provider email network (*n* = 92). An initial broadcast email was sent to CSN on day zero with a reminder email on day 14. On day 28, an initial email was sent to the CureGN provider network, and follow-up email was sent on day 55. The emails included study aims, a link to the website address, the expected length of time to complete the survey, and contact information for questions/concerns (none were received). To prevent repeat sampling from providers affiliated with multiple groups, specific instructions to only complete the survey once were included. Participants were permitted to forward the email to other adult nephrology providers, at their discretion, resulting in snowball sampling. To further increase the sample size, an article describing the study, which included the survey website address, was also published in the March 2018 issue of the American Society of Nephrology Kidney News and posted in their online publication [16].

### 2.3. Statistical Analyses

Demographics, clinical practice characteristics, and frequency of other survey responses were summarized using descriptive statistics (i.e., percentages or frequencies) in univariate analyses. Bivariate analyses, including comparisons of survey responses across provider age (≤45 vs. >45), gender, country of practice (United States vs. Canada), and years of practice (≤10 vs. >10), were performed using Pearson’s chi-squared test or Fisher’s exact test (employed for five or fewer observations). Ordinal (e.g., Likert scale) variables were dichotomized prior to bivariate analyses to assess whether the proportion of confident or very confident (vs. somewhat or not at all confident) responses differed across subgroups. All analyses were performed using Stata, Version 15.1 (Statacorp LLC: College Station, TX, 2017) [17].

## 3. Results

### 3.1. Participant Characteristics and Fellowship Training

In Canada, the survey was sent to 230 nephrologists and 147 nephrology trainees with a response rate of 17%. Ninety-two nephrologists within CureGN were emailed the survey; however, a response rate for the United States (US) could not be computed due to the sampling technique and broader dissemination via Kidney News. Among respondents who initiated the survey, 92.2% completed it fully, with a median time to completion of six minutes. Participants with incomplete surveys were included in the analysis of the questions they answered.

Demographic characteristics of the 154 respondents who submitted the survey are summarized in Table 1. Nephrology training took place in either the US or Canada for 96.8%, and 57.8% were currently practicing in the US. Women made up 52.6% of respondents, and the median age range category was 41–45 years. A majority (77.3%) identified their current practice setting as academic/university, and most (83.1%) answered that their scope of practice included general nephrology. Of those who completed fellowship training, two-thirds (65.5%) reported that they devoted 50% or more of their working time to the clinical care of patients.

Overall, 55.7% of respondents reported that their fellowship program included training in obstetric nephrology and/or women’s health, and this was similar by country of training (*p* = 0.39). Exposure during training primarily came from inpatient (80.5%) and outpatient (59.1%) consults, although over half (57.8%) of respondents also reported formal lectures. Only four participants (2.6%) reported a dedicated clinical rotation in obstetrics. Most (85.1%) saw 10 or fewer pregnant women while in training.

### 3.2. Current Practices

Figure 2 summarizes obstetric history documentation practices during the initial evaluation of a woman of child-bearing age. Only half of nephrologists (51.4%) often/always documented number of prior pregnancies, while even fewer often/always specifically documented previous miscarriages (40.5%) or terminations (33.3%). For hypertensive disorders of pregnancy, 74.3% often/always documented a history of preeclampsia, while 66.2% often/always documented gestational hypertension. Half (51.7%) reported often/always documenting a history of gestational diabetes. With respect to fetal outcomes, one-third (35.4%) often/always documented the gestational age at delivery, and only 17.6% often/always documented the birthweight.

Figure 3 summarizes nephrologists’ practices in (a) contraception/family planning and (b) preconception counseling. In the 12 months prior to taking the survey, 14.5% of nephrologists did not provide contraception/family planning counseling to any woman and, of those that counseled at least one woman, 58.4% counseled less than one woman per month on average. Likewise, 19.2% of providers did not provide preconception counseling to any woman and, of those that counseled at least one, 59.8% counseled less than one woman per month on average. Less than 10% of nephrologists provided counseling to an average of five or more women per month (8.6% on contraception/family planning and 5.3% on preconception). Counseling was more frequent in the US (*p* = 0.004 for contraception/family planning and *p* < 0.001 for preconception). The most common reasons for not providing reproductive or obstetrical counseling were lack of training (53.3%) and little personal knowledge/confidence in the subject area (39.6%). There was also a perceived lack of evidence in the field, cited as a limiting factor by 39% of respondents.

In the 12 months prior to taking the survey, 23.7% did not care for any pregnant women. Of those that cared for at least one pregnant woman, 69% averaged less than one per month. In the past five years, 41% cared for at least one and 8.6% cared for more than five pregnant women with a kidney transplant. In all their years of practice, the majority of nephrologists (71%) reported caring for at least one pregnant woman on dialysis, although only 8.6% cared for more than five pregnant women on dialysis. Among nephrologists that had managed pregnancy in dialysis patients, 98% reported adjusting dialysis intensity. Free text comments were provided by 60.6% of these respondents (63/104) and included prescriptions for dialysis 6 days/week (*n* = 32), daily or nocturnal dialysis (*n* = 19), or targeting 36 h/week (*n* = 4).

Women with CKD who are pregnant or planning pregnancy were reportedly cared for by a variety of providers. Three-fourths of nephrologists (76%) responded that these women were followed by a maternal fetal medicine specialist/high-risk obstetrician. Two-thirds (68.2%) responded that these women were seen by a general or transplant nephrologist, while 44.8% responded that they were seen by a nephrologist specializing in pregnancy. Only 9.2% of nephrologists reported assuming care of pediatric patients (those younger than 18 years old) when they become pregnant. The availability of an interdisciplinary obstetrics and nephrology clinic for pregnant women with CKD was reported by 12.3%.

### 3.3. Confidence in Managing Women’s Health

Figure 4 and Figure 5 show nephrologists’ confidence counseling and managing general women’s health issues (Figure 4) and those related to pregnancy (Figure 5). Outside of pregnancy, very few respondents felt confident/very confident managing menstrual disorders or menopause (6.7% and 6.8%, respectively). Nephrologists felt most confident with the logistics of referring for fertility therapy (34.5% confident/very confident), counseling on contraception (33.1% confident/very confident), and managing osteoporosis (23.7% confident/very confident) though this still represented a minority of respondents. Providers had little confidence discussing surrogacy as an option for patients unable to conceive (90.5% somewhat/not at all confident).

In preconception counseling (Figure 5), nephrologists felt uncomfortable counseling on pregnancy outcomes (only 40.5% confident/very confident) and fetal outcomes (31.8% confident/very confident) according to CKD stage as well as on optimal timing of pregnancy after transplant (31.8% confident/very confident) and in glomerular disease (34.7% confident/very confident). During pregnancy, confidence was low both in managing nephrotic syndrome and in managing dialysis (31.8% confident/very confident and 37.9% confident/very confident, respectively). Hypertension management as it related to women’s health was associated with the highest confidence: 63.5% were confident/very confident managing antihypertensive medications in pregnancy, 62.8% were confident/very confident counseling on the fetotoxicity of antihypertensive medications, and 57.4% were confident/very confident in setting appropriate blood pressure goals in pregnancy. Over half (56.2%) also felt confident/very confident at diagnosing preeclampsia. However, in the postpartum period, less (47.6%) felt confident/very confident managing antihypertensive medications in a breastfeeding mother, and only 37.2% felt confident/very confident managing immunosuppressive medications in a breastfeeding mother.

Table 2 provides comparisons across the subgroups of provider age, gender, country of practice, and years in practice. Female providers and those ≤45 years old were more confident in counseling about contraception (*p* = 0.012 and *p* = 0.043), while those in practice for more than 10 years were more confident in logistically referring for fertility therapy and diagnosing/managing menopause (*p* = 0.040 and *p* = 0.022, respectively). Within pregnancy, nephrologists who were older than 45 years old and those in practice for more than 10 years were more confident in managing dialysis during pregnancy (*p* = 0.030 and *p* = 0.048, respectively). There were no significant differences in confidence based on provider country of practice.

Finally, resources that respondents would find helpful to improve their ability to counsel and manage women with CKD included interdisciplinary guidelines established by obstetrics and nephrology (82.5%), continuing education seminars or case-based materials (66.9%), and access to nephrologists with special interest/training in women’s health (56.5%).

## 4. Discussion

To our knowledge, this is the first study to report findings on nephrologists’ confidence in counseling and managing a spectrum of women’s health issues. Our study identified inconsistent and incomplete obstetric history documentation practices, low rates of contraception and preconception counseling, and variable confidence levels managing women’s health issues among North American nephrologists caring for women with CKD. Although respondents from the US reported more frequent counseling than those in Canada, self-reported confidence was similar across both regions. In addition, over half of all respondents reported that lack of training in women’s health limited their ability to counsel and manage specific issues, echoing findings from a recent national US survey assessing self-perceived training adequacy among recently graduated nephrology fellows, which reported that 45.9% of respondents answered they had “some training, but not enough to feel confident” in renal complications of pregnancy [18]. As such, insufficient women’s health exposure and didactics during both residency and subspecialty training might set the stage for lower confidence and ability in practice. However, 66.9% of our respondents felt that education seminars or case-based materials and round could improve their abilities, revealing a potentially modifiable barrier.

Failure to review a woman’s obstetric history can diminish the quality of a providers’ risk assessment, both for future pregnancy adverse outcomes as well as for future chronic disease development. A history of preeclampsia is not only associated with a two- to four-fold increased risk for chronic hypertension and cardiovascular disease, but may also signify an increased risk for (end-stage renal disease) ESRD [19,20,21,22,23,24]. Gestational diabetes is associated with a seven-fold increased risk for type 2 diabetes and is also a risk factor for future hypertension, cardiovascular disease, and CKD [25,26,27]. Knowledge of the offspring’s birthweight and gestational age can also provide insights into prior placental health and maternal risk for premature cardiac disease and death [28]. Our findings reveal that nephrologists might not be obtaining, or at least not documenting, these important aspects of a woman’s medical history and suggest that more attention should be focused in this area.

Previous studies have shown that women with CKD are hesitant to discuss pregnancy due to fear of judgment and concerns about prioritizing pregnancy over their own health [11]. Thus, it is recommended that physicians initiate the discussion of conception and pregnancy planning early in the disease course of all women of reproductive age [29]. As CKD affects up to 6% of women of childbearing age, our study findings suggest that nephrologists might be failing to counsel a large proportion of women with CKD [30]. Interdisciplinary clinics allow for dedicated comprehensive care and are associated with high patient satisfaction. A pre-pregnancy counseling clinic led by obstetricians and nephrologists in England reported that 90% of surveyed women found the clinic informative, and 89% felt it helped them decide whether to pursue pregnancy [31]. Yet, only 9% of US and 16.9% of Canadian providers in our study reported that their patients were cared for in an interdisciplinary clinic. This highlights development of interdisciplinary care clinics as a potential priority to better serve this high-risk population.

Pregnancy rates in women on dialysis have increased in recent decades. In the Australian and New Zealand Dialysis and Transplant Registry, rates of pregnancies before 1976 were zero, but rose to 3.3 per 1000 person years from 1996–2008 [32]. In our study, most respondents (71%) reported caring for at least one pregnant woman on dialysis and those >45 years old or in practice >10 years felt more confident managing dialysis during pregnancy. Our study also noted the changing practices in dialysis treatment time for pregnant women. A systematic review of case series of dialysis schedules in pregnancy from 2000–2014 found that the most frequently reported schedule was 3.5–4 h per session 5–6 times per week [33]. In our study, 98% (104/106) of nephrologists that had managed pregnancy in dialysis patients reported adjusting dialysis frequency. Of those who specified a regimen, most adjusted frequency to 5–7 days per week, while several utilized nocturnal dialysis or targeted 36 h per week. These practices likely stem from a recent study demonstrating improved live birth rates and longer gestation in women receiving at least 36 h of dialysis per week [34]. Further such multicenter efforts are therefore needed in other areas of women’s health to advance existing knowledge that can in turn be incorporated into evidence-based practice.

There were several limitations to our study. Given the cross-sectional nature of a survey and our reliance on self-reporting, responses were subject to recall bias. Our total sample population was also unknown though the survey was directly emailed to 469 individuals. Our respondents were young, over 50% women, and working in an academic practice setting. Though we did not specifically email private practice individuals, this may signal nonresponse bias from those who did not fill out the survey possibly due to a lack of interest or knowledge in the topic. Within the CSN network, the response rate was 17%, which is comparable to a prior web-based survey of nephrologists regarding management of sex hormone status in women with CKD (21%) [13] and larger than a prior survey of German nephrologists (10%) [14]. To better contextualize our convenience sample, we compared the Canadian survey respondent characteristics to the Canadian Medical Association Nephrologist profile, which contains demographic data of every registered nephrologist in Canada and was updated in August 2018 [35]. Compared to the general Canadian nephrology workforce, Canadian respondents to our survey were more likely to be female (55.4% vs. 39%) and to practice in an academic setting (66% vs. 52%). However, our age distribution (median age 45–54) and percentage of time dedicated to patient care were similar to the nephrology workforce [35].

## 5. Conclusions

In conclusion, our study identifies low frequencies of counseling and low confidence across the spectrum of women’s health. As a broad range of women’s health issues are highly relevant to kidney disease, increased attention to these issues is required to help nephrologists provide patient-centered and disease-specific care. Interventions aimed at increasing the quantity and quality of training and continuous education that nephrologists receive in women’s health—including contraception, menstrual disorders, infertility, menopause, and osteoporosis—appear to be urgently needed. Further, collaboration between obstetricians and nephrologists in establishing interdisciplinary clinics and developing evidence-based guidelines specific to this patient group have emerged as important opportunities to improve the care of women with CKD.

## Figures and Tables

**Figure 1 jcm-08-00176-f001:**
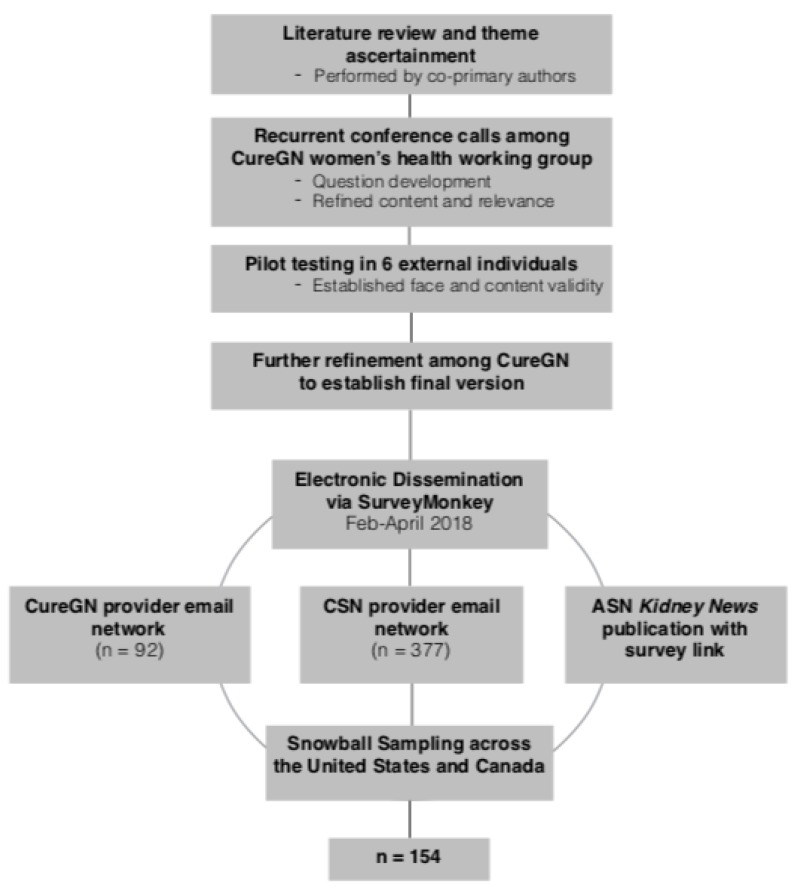
Flowchart of survey design and dissemination.

**Figure 2 jcm-08-00176-f002:**
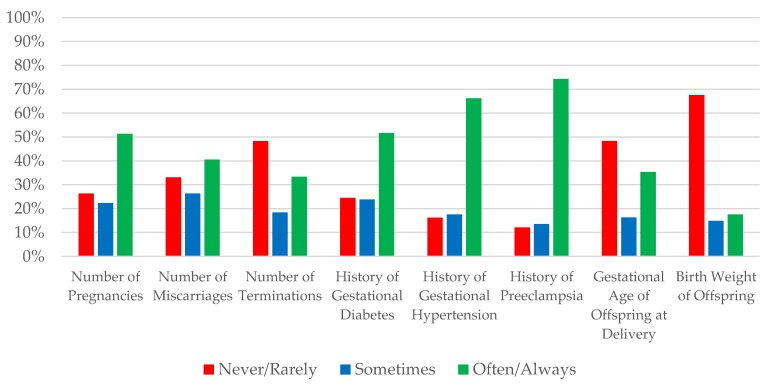
Frequency of obstetrical history documentation by survey respondents.

**Figure 3 jcm-08-00176-f003:**
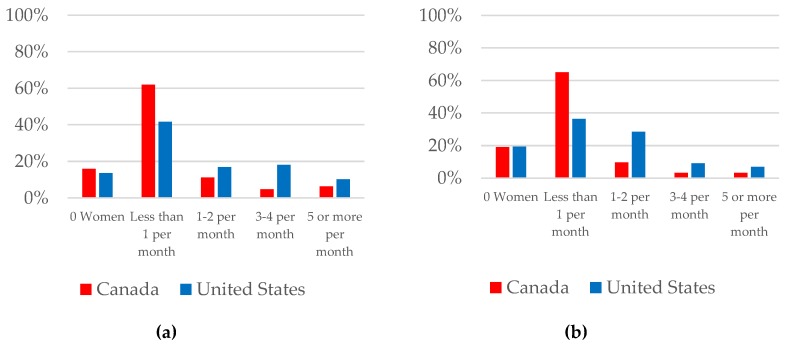
Average frequency of counseling provided by survey respondents in the areas of (**a**) contraception/family planning and (**b**) preconception.

**Figure 4 jcm-08-00176-f004:**
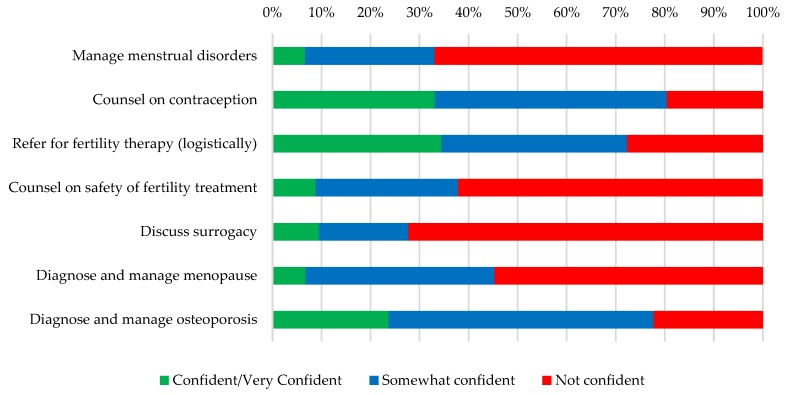
Nephrologists’ confidence managing common women’s health issues.

**Figure 5 jcm-08-00176-f005:**
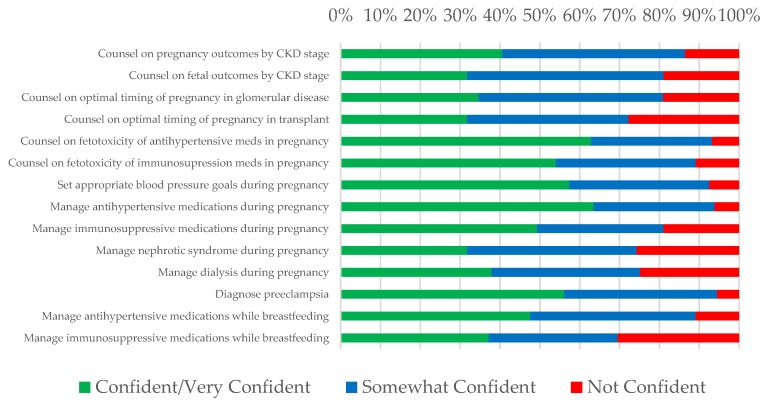
Nephrologists’ confidence managing pregnancy-related topics.

**Table 1 jcm-08-00176-t001:** Survey Respondent Characteristics (*n* = 154).

Respondent Characteristics	Percent of Respondents
Age	
30–35	28.6
36–40	16.9
41–45	18.8
46–50	8.4
51–55	8.4
56–60	7.1
61–65	7.1
Greater than 65	4.6
Gender	
Female	52.6
Male	47.4
Years in Practice	
In training	13.0
0–5	26.0
6–10	16.2
11–15	14.9
16–20	5.8
Greater than 20	24.0
Current Country of Practice	
United States	57.8
Canada	42.2
Current Practice Setting	
Academic/University	77.3
Community	8.4
Hybrid	5.8
In training/Other	8.4
Scope of Practice (select all that apply)	
In training	12.3
CKD/General nephrology	83.8
Dialysis	62.3
Transplant nephrology	33.8
Pediatrics	3.9
Glomerular disease/Other	14.9
Percent Time Devoted to Clinical Care of Patients	
In training	9.7
0–24	14.9
25–49	16.2
50–74	22.7
75–100	36.4

**Table 2 jcm-08-00176-t002:** Percentage of nephrologists who were confident or very confident in counseling or managing women's health issues, by demographic characteristics.

	Age ≤45 years	Age >45 years	*p*	Female	Male	*p*	US	Canada	*p*	Practice ≤10 years	Practice >10 years	*p*
**Manage menstrual disorders**	6.3%	7.6%	0.746 *	10.3%	2.9%	0.103 *	7.9%	5.1%	0.740 *	7.4%	6.0%	1.000 *
**Counsel on contraception**	39.0%	22.6%	0.043	42.3%	22.9%	0.012	37.1%	27.1%	0.207	39.5%	25.4%	0.069
**Refer for fertility therapy (logistically)**	29.5%	43.4%	0.087	33.3%	35.7%	0.761	34.8%	33.9%	0.907	27.2%	43.3%	0.040
**Counsel on safety of fertility treatment**	6.3%	13.2%	0.156	9.0%	8.6%	0.931	7.9%	10.2%	0.628	6.2%	11.9%	0.253 *
**Discuss surrogacy**	6.3%	15.1%	0.080	10.3%	8.6%	0.727	9.0%	10.2%	0.810	6.2%	13.4%	0.163 *
**Diagnose and manage menopause**	4.2%	11.3%	0.168 *	6.4%	7.1%	1.000 *	6.7%	6.8%	1.000 *	2.5%	11.9%	0.043 *
**Diagnose and manage osteoporosis**	21.1%	28.3%	0.320	20.5%	27.1%	0.343	21.4%	27.1%	0.419	19.8%	28.4%	0.220
**Counsel on pregnancy outcomes by CKD stage**	39.0%	43.4%	0.597	41.0%	40.0%	0.899	37.1%	45.8%	0.292	37.0%	44.8%	0.340
**Counsel on fetal outcomes by CKD stage**	31.6%	32.1%	0.950	35.9%	27.1%	0.253	31.5%	32.2%	0.924	29.6%	34.3%	0.541
**Counsel on optimal timing of pregnancy in glomerular disease**	36.2%	32.1%	0.617	37.7%	31.4%	0.428	37.5%	30.5%	0.383	33.8%	35.8%	0.793
**Counsel on optimal timing of pregnancy after transplant**	33.7%	28.3%	0.500	32.1%	31.4%	0.935	34.8%	27.1%	0.324	30.9%	32.8%	0.798
**Counsel on fetotoxicity of antihypertensive medications**	65.3%	58.5%	0.414	62.8%	62.9%	0.996	66.3%	57.6%	0.285	63.0%	62.7%	0.972
**Counsel on fetotoxicity of immunosuppressive medications**	54.7%	52.8%	0.823	56.4%	51.4%	0.544	59.6%	45.8%	0.099	53.1%	55.2%	0.795
**Set appropriate blood pressure goals during pregnancy**	59.0%	54.7%	0.618	52.6%	62.9%	0.206	56.2%	59.3%	0.705	55.6%	59.7%	0.612
**Manage antihypertensive medications during pregnancy**	65.3%	60.4%	0.554	59.0%	68.6%	0.226	64.0%	62.7%	0.869	63.0%	64.2%	0.878
**Manage immunosuppressive medications during pregnancy**	49.5%	49.1%	0.961	46.2%	52.9%	0.415	53.9%	42.4%	0.168	46.9%	52.2%	0.519
**Manage nephrotic syndrome during pregnancy**	31.6%	32.1%	0.950	29.5%	34.3%	0.531	25.4%	36.0%	0.178	30.9%	32.8%	0.798
**Manage dialysis during pregnancy**	31.6%	50.0%	0.030	31.2%	45.6%	0.074	36.0%	41.1%	0.536	30.9%	46.9%	0.048
**Diagnose preeclampsia**	57.5%	53.9%	0.675	53.9%	58.8%	0.545	55.2%	57.6%	0.769	52.5%	60.6%	0.326
**Manage antihypertensive medications during breastfeeding**	51.1%	41.5%	0.265	47.4%	47.8%	0.962	51.1%	42.4%	0.297	51.3%	43.3%	0.335
**Manage immunosuppressive medications during breastfeeding**	34.7%	41.5%	0.414	33.3%	41.4%	0.309	40.5%	32.2%	0.309	32.1%	43.3%	0.161

* Fishers exact test used rather than Pearson’s chi-squared test due to less than five observations.

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
