# Peer review of "Confidence in Women’s Health: A Cross Border Survey of Adult Nephrologists"

_jcm, 2019, doi:10.3390/jcm8020176_

Reviewer 1 Report

This manuscript highlights an important deficiency in the clinical nephrology community--that of counseling and managing women's health issues. This survey-based study was well-conceived and provides illuminating, informative data. The manuscript is well-written, and the data is thoroughly analyzed and clearly presented. Furthermore, the authors identify possible improvements to nephrology training curricula that could improve providers' ability to manage women's health issues in their CKD patients. The authors describe how provider gender (female vs. male) and current country of practice (US vs. Canada) affect provider confidence. I'm also curious as to how provider age and/or years in practice affect provider confidence and/or practice patterns--could this also be assessed? 

Author Response

Dear Reviewer

Thank you for your kind comments and excellent suggestion. We added a table that describes confidence by age, gender, country of practice and years of practice. Please see Table 2 for the response to your request. 

This text was also added to the manuscript:

Female providers and those << span="">45 years old were more confident in counseling about contraception (p= 0.012 and p= 0.043) while those in practice more than 10 years were more confident in logistically referring for fertility therapy and diagnosing/managing menopause (p= 0.040 and p= 0.022, respectively).  Within pregnancy, nephrologists who were older than 45 years old and those in practice more than 10 years were more confident in managing dialysis during pregnancy (p= 0.030 and p= 0.048, respectively). There were no significant differences in confidence based on provider country of practice.

Reviewer 2 Report

The authors highlighted an important area in Nephrology training that needs more attention. Major Nephrology textbooks such as comprehensive clinical nephrology have chapters dedicated to " Pregnancy and Renal disease", so it is very reasonable to include continuing education seminars or case based conferences for Nephrology fellows during their morning report or noon conference time, to improve confidence in taking care of women with CKD/ESRD.

Author Response

Dear Reviewer,

We have included a statement to note the potential teaching venues as suggested.

The text reads as follows:

However, 66.9% of our respondents felt that education seminars or case-based materials and rounds could improve their abilities, revealing a potentially modifiable barrier.

Reviewer 3 Report

This is a great study which stresses the importance of Obstetrics training/basics for Nephrology training. Also, the need for collaboration between obstetricians and nephrology. 

I feel the low confidence cannot be just attributed to nephrology fellows, it likely may be the same for internal medicine trained residents. So there may be a need for more training right from residency.

Knowledge of Obstetrics/women's health issues among nephrologists. It is not studied in the past. It adds the importance of the need for the nephrologists to know about obstetrics issues.It is relevant and important Paper is well written. Conclusions consistent with the arguments.
 They address the question posed.

Author Response

Dear Reviewer,

Thank you for your comments. We added a statement to note that this training should start as early as residency. The text reads as follows:

As such, insufficient women’s health exposure and didactics during training, during both residency and subspecialty training, might set the stage for lower confidence and ability in practice.  However, 66.9% of our respondents felt that education seminars or case-based materials and round could improve their abilities, revealing a potentially modifiable barrier.